# Learning Over Molecular Conformer Ensembles: Datasets and Benchmarks

## Abstract

Molecular Representation Learning (MRL) has proven impactful in numerous biochemical applications such as drug discovery and enzyme design. While Graph Neural Networks (GNNs) are effective at learning molecular representations from a 2D molecular graph or a single 3D structure, existing works often overlook the flexible nature of molecules, which continuously interconvert across conformations via chemical bond rotations and minor vibrational perturbations. To better account for molecular flexibility, some recent works formulate MRL as an ensemble learning problem, focusing on explicitly learning from a set of conformer structures. However, most of these studies have limited datasets, tasks, and models. In this work, we introduce the first MoleculAR Conformer Ensemble Learning (MARCEL) benchmark to thoroughly evaluate the potential of learning on conformer ensembles and suggest promising research directions. MARCEL includes four datasets covering diverse molecule- and reaction-level properties of chemically diverse molecules including organocatalysts and transition-metal catalysts, extending beyond the scope of common GNN benchmarks that are confined to drug-like molecules. In addition, we conduct a comprehensive empirical study, which benchmarks representative 1D, 2D, and 3D molecular representation learning models, along with two strategies that explicitly incorporate conformer ensembles into 3D MRL models. Our findings reveal that direct learning from an accessible conformer space can improve performance on a variety of tasks and models.

## 1 Introduction

Recent years have seen the emergence of Molecular Representation Learning (MRL) as a promising approach for modeling molecules with machine learning. In the typical formulation, MRL maps discrete molecular objects to continuous features in a data-driven manner, encoding complex chemical structures into representation vectors that can subsequently be utilized in different downstream tasks. In particular, MRL now underpins a variety of biochemical applications spanning molecular property prediction to the design of novel drug candidates [1–3].

Traditional approaches often encode chemical compounds with fingerprints, such as extended-connectivity fingerprints [4, 5], which indicate the existence of certain substructures as binary bits in a fixed-length sequence. Such line-based representations are concise and efficient, but have limited expressive power and have difficulty in capturing 3D structural information such as bonding geometries and global shapes, which can be important for analyzing molecular properties and chemical reactivity [6, 7]. Recently, Graph Neural Networks (GNNs) have become an increasingly popular method of learning molecular representations by treating molecules as graph-structured objects. Existing GNN models for MRL can be broadly classified into two categories: 2D topological models [8–11] and 3D geometric models [12–17]. 2D GNNs typically model the molecular connectivity as a flat 2D graph with atoms as nodes and bonds as edges, learning representations of chemical environments by iteratively passing messages between neighboring atoms. Although powerful in the absence of

Submitted to NeurIPS 2021 AI for Science Workshop.

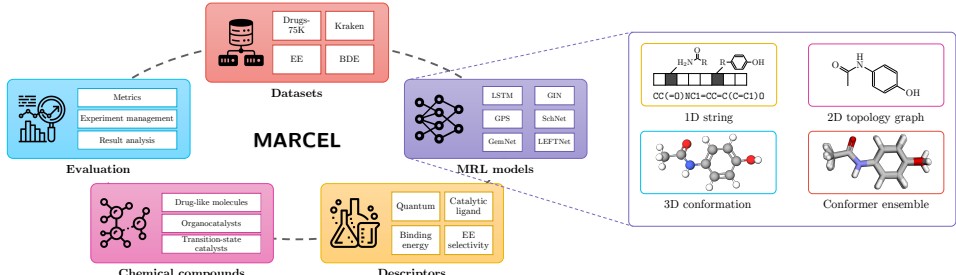

Figure 1: We present a MARCEL benchmark that comprehensively evaluates the potential of learning on conformer ensembles across a diverse set of molecules, datasets, and models.

structural information, 2D GNNs may fail to capture key conformational effects or stereochemical properties like chirality [18, 19], which is critical for modeling molecular interactions in areas such as drug design or chemical catalysis. Conversely, 3D GNNs are designed to model molecular conformers (conformations), which describe the structure of molecules in 3D space. Thus, these models have found widespread adoption for modeling electronic properties, predicting conformer energies and forces, and scoring interactions between ligands and proteins, amongst other applications.

In almost all applications, benchmarks, and demonstrations, 3D GNN models focus on encoding *individual* conformer structures. It is critical to recognize that in reality molecules are not rigid, static objects; rather, thermodynamically-permissible rotations of chemical bonds, small vibrational motions, and dynamic intermolecular interactions cause molecules to continuously convert between different conformations [20]. As a consequence, many experimentally observable chemical properties depend on the full distribution of thermodynamically-accessible conformers. For example, a molecule needs to be arranged into a particular pose to bind to a target protein, and this binding conformation changes depending on the dynamic interaction between the molecule and the target [21]. Also, it is often challenging to determine *a priori* the conformers that predominantly contribute to molecular properties without doing prohibitively expensive simulations. Therefore, a natural question arises: can we leverage the *collective* power of many different conformer structures lying on the local minima of the potential energy surface, also known as the *conformer ensemble*, to improve MRL models?

As shown by the empirical evidence from various studies, learning from an explicit conformer ensemble can prove to be advantageous for many tasks, including property and energy prediction [22–24], key conformer pose identification [25], and RNA sequence design [26]. However, these studies have been mostly confined to small-scale datasets, a limited set of tasks, and a restricted set of model architectures. As a result, it remains unclear (1) to what extent 2D GNNs can implicitly model molecular flexibility and (2) whether the *explicit* encoding of conformer ensembles can improve the performance of 3D models that traditionally encode only one single conformer.

In this paper, we present the first MoleculAR Conformer Ensemble Learning (MARCEL) benchmark. As shown in Figure 1, MARCEL covers a diverse range of chemical space, which focuses on four chemically-relevant tasks for both molecules and reactions, with an emphasis on Boltzmann-averaged properties of conformer ensembles computed at the Density-Functional Theory (DFT) level. Our datasets encompass a variety of compounds with high-quality conformers, including organocatalysts and transition-metal catalysts, extending beyond the scope of conventional GNN benchmarks which are often restricted to drug-like molecules. Moreover, we implement a comprehensive benchmark suite that enables extensive empirical studies across representative 1D, 2D, and 3D MRL models. We further explore the advantages of leveraging conformer ensembles through two straightforward strategies: (1) augmenting training samples by randomly selecting one conformer from the ensemble for each molecule and (2) applying an explicit multi-instance ensemble learning layer, which aggregates individual conformer embeddings.

Our experimental results confirm the potential effectiveness of incorporating conformer ensembles in MRL, highlighting the improvements over conventional single-conformation 3D networks. However, it is important to understand the heterogeneity of outcomes based on different dataset characteristics, task objectives, and model choices. Our investigation yields three key findings: (1) Leveraging molecular conformers by incorporating explicit set encoders, as a part of conformer ensemble learning strategies, can improve single-conformer 3D MRL models performance. (2) Data augmentation through conformer sampling may offer potential benefits, evidenced by improved results in the BDE dataset, suggesting a method to increase model robustness against imprecise structures. (3) Model selection for MRL depends on dataset sizes and tasks, with traditional 1D fingerprints and 2D models preferred for smaller datasets and 3D models for larger or reaction-focused tasks.

## 2 Problem Formulation

We represent a 2D molecular graph as a tuple $\mathsf{G} = (\mathcal{V}, \mathcal{E}, \boldsymbol{X}, \boldsymbol{W})$, where $\mathcal{V} = \{v_i\}_{i=1}^{|\mathcal{V}|}$ is the node set with each node corresponding to an atom, and $\mathcal{E} \subseteq \mathcal{V} \times \mathcal{V}$ is the edge set representing chemical bonds as edges between nodes. Further, $\boldsymbol{X} \in \mathbb{R}^{d_v \times |\mathcal{V}|}$ contains vector attributes for each node, and $\boldsymbol{W} \in \mathbb{R}^{d_w \times |\mathcal{E}|}$ contains attributes for each edge. When modeling chemical reactions, we represent a molecule-molecule complex as a pair of graphs $(\mathsf{G}_1, \mathsf{G}_2)$. In this case, the conformation describes the combined structure of the interacting molecules. For a given molecule or molecular complex, we assume that its geometry can be effectively characterized by a representative set of discrete, sampled conformers from the thermodynamically-accessible conformer distribution. Formally, this set can be denoted as $\mathcal{C} = \{\boldsymbol{C}_i\}_{i=1}^{|\mathcal{C}|}$, where $\boldsymbol{C}_i \in \mathbb{R}^{|\mathcal{V}| \times 3}$ represents one conformer structure in 3D space. In reality, the conformer distribution is continuous; $\mathcal{C}$ in our study contains representative samples of the infinite set. Each conformer in the sampled ensemble is associated with a statistical weight given by $p_i = \dfrac{\exp\left(-\frac{e_i}{k_B T}\right)}{\sum_j \exp\left(-\frac{e_j}{k_B T}\right)}$, which corresponds to its probability under experimental conditions. Here, $e_i$ is the energy of the conformer $\boldsymbol{C}_i$, $k_B$ is the Boltzmann constant, and $T$ is the temperature. Notably, $p_i$ is not prior information to the models analyzed in this benchmark. Rather, we use a discrete approximation of $p_i$ to compute the ground-truth labels for our regression tasks.

## 3 Datasets and Tasks

MARCEL contains four small-to-large-scale datasets involving nine regression tasks with considerably diverse chemistry. Drugs-75K and Kraken focus on molecular properties, while EE and BDE focus on reaction-centric properties. MARCEL includes molecules with high structural flexibility, evidenced by an average number of rotatable bonds exceeding 5. Table 1 summarizes the datasets.

**Drugs-75K** is a subset of the GEOM-Drugs [27] dataset, which includes 75,099 molecules with at least 5 rotatable bonds. For each molecule, we focus on three important quantum chemical descriptors: ionization potential, electron affinity, and electronegativity [28]. The tasks are to predict the Boltzmann-averaged value of each property across the conformer ensemble $\langle y \rangle_{k_B} = \sum_{\boldsymbol{C}_i \in \mathcal{C}} p_i y_i$, where $y_i$ is a conformer-specific property. We are given each $\boldsymbol{C}_i$, and the goal is to predict $\langle y \rangle_{k_B}$ from the molecular graph $\mathsf{G}$, a single conformer $\boldsymbol{C}_i \in \mathcal{C}$, or the set $\mathcal{C}$.

**Kraken** [29] is a dataset of 1,552 monodentate organophosphorus (III) ligands along with their DFT-computed conformer ensembles. In this study, we consider four 3D ligand descriptors exhibiting significant variance among conformers: Sterimol $B_5$, Sterimol L, buried Sterimol $B_5$, and buried Sterimol L. These descriptors quantify the steric features of each ligand in units of Å and are often employed for Quantitative Structure-Activity Relationship (QSAR) modeling in catalyst design.

As in the Drugs-75K tasks, the goal is to predict the Boltzmann-averaged value of each property across the conformer ensemble from the molecular graph $\mathsf{G}$, a single conformer $\boldsymbol{C}_i \in \mathcal{C}$, or the set $\mathcal{C}$.

**EE** [30] is a dataset of 872 catalyst-substrate pairs involving 253 Rh-bound atropisomeric catalysts derived from chiral bisphosphine, with 10 enamides as substrates. The dataset includes conformations of catalyst-substrate transition state complexes in two separate pro-S and pro-R configurations. The task is to predict the Enantiomeric Excess (EE) of the chemical reaction involving the substrate. Unlike properties in Drugs-75K and Kraken, EE depends on the conformer ensembles of *each* pro-R and pro-S complex. The goal is to predict EE from the graphs of the catalyst and substrate $(\mathsf{G}_{\text{cat}}, \mathsf{G}_{\text{sub}})$, a conformer $\boldsymbol{C}_i^{(\text{R})} \in \mathcal{C}^{(\text{R})}$ and $\boldsymbol{C}_i^{(\text{S})} \in \mathcal{C}^{(\text{S})}$ for each complex, or the ensembles $\mathcal{C}^{(\text{R})}$ and $\mathcal{C}^{(\text{S})}$.

**BDE** [31] is a dataset containing 5,915 organometallic catalysts $\text{ML}_1\text{L}_2$ consisting of a metal center coordinated to two flexible organic ligands. The data includes conformations of each unbound catalyst, as well as conformations of the catalyst when bound to ethylene and bromide after oxidative addition with vinyl bromide. Each catalyst has an electronic binding energy to be predicted. Although the binding energies are computed via DFT, the conformers provided for modeling are initially generated with Open Babel [32] followed by further geometry optimization, which ensures that the 3D structures are likely to be the global minimum energy conformers at the force field level [31]. This dataset realistically represents the setting in which precise conformer ensembles are unknown at inference. The task is to predict the binding energy from the graphs of the unbound and bound catalyst, sampled conformers $\boldsymbol{C}_i^{(\text{unbound})} \in \mathcal{C}^{(\text{unbound})}$ and $\boldsymbol{C}_i^{(\text{bound})} \in \mathcal{C}^{(\text{bound})}$, or the ensembles $\mathcal{C}^{(\text{unbound})}$ and $\mathcal{C}^{(\text{bound})}$.

Table 1: Statistics of the four datasets. The numbers of heavy atoms and rotatable bonds ("rot. bonds") are averaged per conformer.

| Dataset | # Molecules | # Conformers | # Heavy atoms | # Rot. bonds | # Targets | Atomic species |
|---------|-------------|--------------|---------------|--------------|-----------|----------------|
| Drugs-75K | 75,099 | 558,002 | 30.56 | 7.53 | 3 | H, C, N, O, F, Si, P, S, Cl |
| Kraken | 1,552 | 21,287 | 23.70 | 9.05 | 4 | H, B, C, N, O, F, Si, P, S, Cl, Fe, Se, Br, Sn, I |

| Dataset | # Reactions | # Conformers | # Heavy atoms | # Rot. bonds | # Targets | Atomic species |
|---------|-------------|--------------|---------------|--------------|-----------|----------------|
| EE | 872 | Pro-R: 14,807 Pro-S: 13,999 | 59.32 | 18.57 | 1 | H, C, N, O, F, P, Cl, Br, Rh |
| BDE | 5,915 | Ligand: 73,834 Complex: 40,264 | 29.62 32.38 | 6.99 6.99 | 1 | H, C, N, O, F, P, Cl, Ni, Cu, Br, Pd, Ag, Pt, Au |

**Dataset Preparation.** We implement several preprocessing steps to ensure the quality and validity of our datasets and facilitate their integration into machine learning models.

- **Conformer deduplication.** To eliminate redundant conformers in each ensemble $\mathcal{C}$, we first align every pair of conformers using RDKit [33], accounting for symmetric atom permutations. Subsequently, we employ Butina clustering [34] based on the Root Mean Square Deviation (RMSD) values derived from conformer alignment. Within each cluster, we select the conformer with the lowest energy. Note that Boltzmann-averaged regression labels are computed *before* deduplication.

- **Selection of molecules.** We focus on modeling flexible molecules, for which conformer ensemble learning may be relevant to capture their properties. Hence, we only retain molecules with more than 5 rotatable bonds. We also remove molecules with missing 3D geometries or 2D graphs.

## 4    Benchmarking Molecular Representation Learning Models

The representation of molecular data is crucial for applying machine learning models to problems in chemistry and biology. These representations typically include 1D strings, 2D topological graphs, and 3D geometric graphs. For a comprehensive benchmark for MRL models, our MARCEL includes a diverse representative selection of models for each of the aforementioned molecular representations. In this section, we provide an overview of these models and describe how they are tailored to our tasks. We also introduce two strategies of explicitly encoding conformer ensembles using 3D models.

### 4.1    1D Models

Our 1D baselines include Random Forest [35] models operating on molecular fingerprints [33, 36, 37]. Fingerprints convert a molecular graph into a bit array indicating the presence of chemical substructures and are widely used for cheminformatics and QSAR modeling in the low-data regime. Additionally, we include Long Short-Term Memory (LSTM) [38] and Transformer [39] models, popular sequence-based neural network architectures, operating on SMILES strings. For the BDE and EE datasets, we concatenate the SMILES of each molecule or complex with a "." symbol and use a single sequence encoder. Further details on model implementations can be found in Appendix B.1.

### 4.2    2D Graph Neural Networks

We employ four widely-used GNN models as 2D baseline methods, including Graph Isomorphism Network (GIN) [40], GIN with Virtual Node (GIN-VN) [41], ChemProp [42], and GraphGPS [43]. Following OGB protocols [41], we employ a diverse set of atomic features such as aromaticity and hybridization for nodes, as well as bond features like ring information for edges (Appendix B.2). For the EE and BDE datasets, we employ a two-tower architecture with two separate 2D GNN models: for EE, since both pro-S and pro-R complexes share the same 2D graph, we leverage two separate GNNs to encode the catalyst and substrate; for BDE, we also encode the unbound and bound catalysts separately. We then concatenate these together to obtain the system-level embeddings.

### 4.3    3D Graph Neural Networks

We include six representative 3D GNNs that encompass diverse modeling perspectives. For invariant networks, our experiments involve SchNet [12], DimeNet++ [13], and GemNet [14]. For equivariant networks, we include PaiNN [15], ClofNet [16], and LEFTNet [17].

We use atom types as the sole atom features for the 3D models. For both training and inference on Drug-75K, Kraken, and EE datasets, all the single-conformer 3D models encode the lowest-energy conformer of each conformer ensemble, which has the largest Boltzmann weight and hence provides the strongest model. Since imprecise conformers are encoded for the BDE task, we use a fixed, randomly sampled conformer for each unbound- and bound-catalyst during training and inference.

The 3D models also employ a two-tower architecture for the EE and BDE datasets. Two separate 3D GNNs are used to encode representations for each pro-S and pro-R complex in EE, or for each catalyst and bound complex in BDE, which are then concatenated to form the final representations.

We note that although using the lowest-energy conformer will yield the strongest performance, this setting can be unrealistic: it is often not possible to identify the lowest energy conformer without searching the entire conformer space. The lowest energy conformer can also depend on the force field used for geometry optimization, which may neglect experimental conditions such as solvents.

## 4.4 Incorporating Conformer Ensembles into Molecular Representations

3D geometric models primarily focus on learning representations from individual 3D structures. Although some models preserve global symmetries such as SE(3)-equivariance, these models do not learn representations that capture conformational flexibility which is caused by internal degrees of freedom such as bond rotations. Here, we describe two straightforward strategies that model conformational flexibility by explicitly leveraging conformer ensembles.

### 4.4.1 Strategy 1: Training-Time Data Augmentation via Conformer Sampling

A direct approach to modeling conformer flexibility is to simply enrich the training data by randomly sampling a conformer from the ensemble during each training epoch. Formally, for a given molecule G and its conformer ensemble $\mathcal{C}$, we randomly select a conformer with uniform probability $p = 1/|\mathcal{C}|$ while using the same training label for each conformer. Note that during inference, the lowest-energy conformer is used to evaluate the model performance. This strategy aligns with learning representations invariant to conformational changes, thus implicitly encoding the flexibility of molecular structures, and has been shown to be useful for learning chirality-sensitive 3D representations [19]. When conformer ensembles are available, the strategy is computationally efficient as it maintains the same complexity as the base 3D model. Unlike the other ensemble methods, this strategy can be used if conformer ensembles are only available at training time. In Appendix C, we evaluate two alternative scenarios where conformer ensembles are also available during evaluation.

### 4.4.2 Strategy 2: Ensemble Learning with Explicit Set Encoders

The second strategy utilizes a set encoder to simultaneously encode the entire conformer ensemble $\mathcal{C}$ at both training and inference time. Inspired by the multi-instance learning framework [44–46], this strategy first employs 3D GNNs to generate individual conformer embeddings and then aggregates these embeddings using a set encoder, as illustrated in Figure 2.

Formally, for each conformer $C_i \in \mathcal{C}$, we obtain its corresponding embedding $z_i = f(G, C_i) \in \mathbb{R}^d$, where $f$ is a single-conformer 3D model and $d$ is the embedding dimension. Note that the embedding $z$ is a (3D) graph-level representation resulting from a pooling function over the node-level embeddings after message passing. To further aggregate these embeddings $\mathcal{Z} = \{z_i\}_{i=1}^{|\mathcal{C}|}$ into a single molecular representation, we consider the following three set encoders:

- **Mean pooling** simply computes the mean of all the conformer embeddings.

- **DeepSets** [47] utilizes a permutation-invariant function to process a set of inputs. It first applies a MultiLayer Perceptron (MLP) $h$ to each conformer embedding and then aggregates the transformed embeddings using sum pooling followed by another MLP $g$:

$$s^{\text{DS}} = g\left(\sum_{i=1}^{|\mathcal{C}|} h(z_i)\right). \tag{1}$$

This method retains more discernible information from individual embeddings compared to mean pooling at a cost of two non-linear functions.

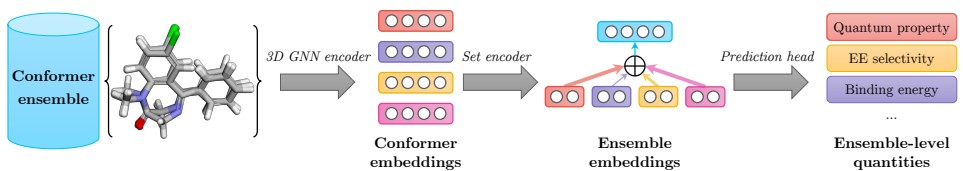

Figure 2: Conformer ensemble learning with explicit set encoders (Strategy 2). Individual conformer embeddings are first obtained via 3D GNN encoders. Then, a set encoder is employed to aggregate conformer embeddings. Finally, a linear projection head is used to generate the prediction.

- **Self-attention** [48] further computes a weighted sum of the embeddings, where the weights are obtained by applying a softmax function to the dot product of the embeddings:

$$s^{\text{ATT}} = \sum_{i=1}^{|\mathcal{C}|} c_i, \quad \text{where} \; c_i = g\left(\sum_{j=1}^{|\mathcal{C}|} \alpha_{ij} h(z_j)\right), \; \alpha_{ij} = \frac{\exp((Wh(z_i))^\top (Wh(z_j)))}{\sum_{k=1}^{|\mathcal{C}|} \exp((Wh(z_i))^\top (Wh(z_k)))}. \tag{2}$$

Here, $W \in \mathbb{R}^{d \times d}$ is a learnable weight matrix. This approach can capture conformer interactions.

By employing these set encoders, we can learn a model that is more sensitive to the full range of conformer variations present in the ensemble. After obtaining the ensemble embeddings, we further apply a linear projection head to generate the final prediction.

## 5 Experiments

### 5.1 Experimental Configurations

Each dataset is partitioned randomly into three subsets: 70% for training, 10% for validation, and 20% for test. Each model is trained over 2,000 epochs using the Adam optimizer [49] with early stopping triggered if there is no improvement on the training loss over 200 epochs. For all nine regression targets, experiments are repeated three times, and the results reported correspond to the model that performs best on the validation set in terms of Mean Absolute Error (MAE).

The Boltzmann-averaged targets are computed over all available conformers. For ensemble learning models, we cap the number of encoded conformers per molecule to a maximum of 20, which empirically improves training stability and leads to better performance. To ensure a fair comparison, the hidden dimension size is uniformly set to 128 for all models. Other settings mostly follow the original configurations as described in the respective papers. We specify all hyperparameters and describe experimental environments in Appendix B.3.

### 5.2 Results and Analysis

We summarize the performance of the 1D, 2D, and 3D MRL models in Table 2. Figure 3 reports the *performance changes* in Mean Absolute Error (MAE) for each 3D model when applying the ensemble learning strategies. The raw performance data with standard deviation and the parameter size of each model can be found in Appendix D. In summary, although performance varies across the datasets, tasks, and models, the ensemble learning strategies improve upon 3D models that only encode one conformer in 48 out of 54 experiments across 9 tasks and 6 base models, demonstrating the effectiveness of conformer ensemble learning. Our analysis leads to the following key observations.

**Observation 1. The conformer ensemble learning strategy with explicit set encoders frequently yields improved performance.**

Figure 3 indicates that encoding conformer ensembles can substantially reduce test error, achieving improvements in 108 experiments across all 9 tasks, 6 base models, and 3 set encoders, most notably on the tasks in the smaller-sized Kraken dataset. This, however, does not always extend to larger datasets like Drugs-75K. We conjecture that for Drugs-75K, the computational burden of encoding all conformers in each ensemble alters the learning dynamics of the underlying model, making training more challenging. A similar finding was reported by Axelrod and Gómez-Bombarelli [23].

Among the three set encoders, DeepSets consistently demonstrates significant improvements in 42 out of 54 experiments across 9 tasks and 6 base 3D models. We conjecture that this superior performance

Table 2: Performance of 1D, 2D, and 3D baseline MRL models and the best results from ensemble learning strategies on 3D GNNs. The metric used is the Mean Absolute Error (MAE, ↓). The **bold** indicates the best-performing model, while underlined denotes the second-best.

| Category | Model | Drugs-75K | | | Kraken | | | | EE | BDE |
| | | IP | EA | $\chi$ | $B_5$ | L | $BurB_5$ | BurL | | |
| --- | --- | --- | --- | --- | --- | --- | --- | --- | --- | --- |
| | Random forest | 0.4987 | 0.4747 | 0.2732 | 0.4760 | 0.4303 | 0.2758 | 0.1521 | 61.2963 | 3.0335 |
| 1D | LSTM | 0.4788 | 0.4648 | 0.2505 | 0.4879 | 0.5142 | 0.2813 | 0.1924 | 64.0088 | 2.8279 |
| | Transformer | 0.6617 | 0.5850 | 0.4073 | 0.9611 | 0.8389 | 0.4929 | 0.2781 | 62.0816 | 10.0771 |
| | GIN | 0.4354 | 0.4169 | 0.2260 | 0.3128 | 0.4003 | 0.1719 | 0.1200 | 62.3065 | 2.6368 |
| 2D | GIN+VN | 0.4361 | 0.4169 | 0.2267 | 0.3567 | 0.4344 | 0.2422 | 0.1741 | 62.3815 | 2.7417 |
| | ChemProp | 0.4595 | 0.4417 | 0.2441 | 0.4850 | 0.5452 | 0.3002 | 0.1948 | 61.0336 | 2.6616 |
| | GraphGPS | 0.4351 | 0.4085 | 0.2212 | 0.3450 | 0.4363 | 0.2066 | 0.1500 | 61.6251 | 2.4827 |
| | SchNet | 0.4394 | 0.4207 | 0.2243 | 0.3293 | 0.5458 | 0.2295 | 0.1861 | 17.7421 | 2.5488 |
| | DimeNet++ | 0.4441 | 0.4233 | 0.2436 | 0.3510 | 0.4174 | 0.2097 | 0.1526 | 14.6414 | **1.4503** |
| 3D | GemNet | 0.4069 | 0.3922 | **0.1970** | 0.2789 | 0.3754 | 0.1782 | 0.1635 | 18.0338 | 1.6530 |
| | PaiNN | 0.4505 | 0.4495 | 0.2324 | 0.3443 | 0.4471 | 0.2395 | 0.1673 | 20.2359 | 2.1261 |
| | ClofNet | 0.4393 | 0.4251 | 0.2378 | 0.4873 | 0.6417 | 0.2884 | 0.2529 | 33.9473 | 2.6057 |
| | LEFTNet | 0.4174 | 0.3964 | 0.2083 | 0.3072 | 0.4493 | 0.2176 | 0.1486 | 19.7974 | 1.5328 |

is due to its ability of effectively modeling set objects at a relatively minor computational overhead of two non-linear transformations. On the other hand, the simple mean pooling approach loses discriminative power across the conformers in the ensemble, resulting in inferior performance. It is also noteworthy that the attention models exhibit mixed results compared to the vanilla 3D models, despite theoretically being the most powerful set encoders. This inconsistency might be attributable to the computational intricacy of the self-attention layer, which models the pairwise relationship among conformers in each ensemble and hence could require more sophisticated training strategies. Future research should consider developing better neural architectures that are specifically designed to more efficiently encode structural information from conformer ensembles.

**Observation 2. Sampling conformers at training time can improve performance, especially on imprecise conformer structures.**

We observe that data augmentation improves performance on 34 experiments, especially on the challenging BDE dataset, where the other ensemble learning strategies often do not help. Note that the conformers in the BDE dataset originate from Open Babel, as opposed to the golden-standard DFT-level conformers present in other datasets. This suggests that training with randomly sampled conformers might offer robustness to noise in the imprecise structures. On other tasks, randomly sampling the conformers at each epoch may help the model learn an invariance to conformational changes, but does not always increase performance for all 3D models. This might be because the sampling probability is uniform across the entire conformer set, which does not respect the underlying Boltzmann weight of each conformer. In future work, it may be worthwhile to investigate whether more physics-informed sampling strategies could lead to more consistent performance gains.

**Observation 3. No model consistently outperforms the rest, with substantial task dependencies.**

The results in Table 2 suggest that no single model outperforms the others across all tasks. Of the 1D models, LSTM outperforms Random Forest and Transformer models on Drugs-75K and BDE, demonstrating the effectiveness of SMILES-based representations of molecules on large-scale datasets. For small datasets such as Kraken and EE, Random Forests outperform sequence models at a lower computational cost, indicating that traditional models are superior in the low-data regime.

Amongst 2D models, GIN delivers the best performance on four tasks compared to all other models; GraphGPS also demonstrates strong performance on several tasks ($B_5$, L, and BurL). Surprisingly, the 2D models are also competitive with some 3D models on the large-scale Drugs-75K tasks. This is possibly due to the fact that the electronic properties in Drugs-75K are not as sensitive to conformational changes, thus explicitly modeling the structures in 3D may not be necessary. However, all 2D models perform worse as compared to the 3D models in the reaction datasets EE and BDE, indicating the important role of spatial interactions in determining reaction-related properties.

For 3D models, GemNet and LEFTNet excel in IP, EA, and $\chi$. The complexity of these two equivariant models may especially benefit from the large dataset size of Drugs-75K. For Kraken and the two reaction datasets, DimeNet++ — an invariant model — achieves promising performance, suggesting that highly-complex 3D models may be less useful for chemical applications with small-to-medium sized datasets. On EE, we observe that 3D models remarkably outperform 1D and 2D models, likely

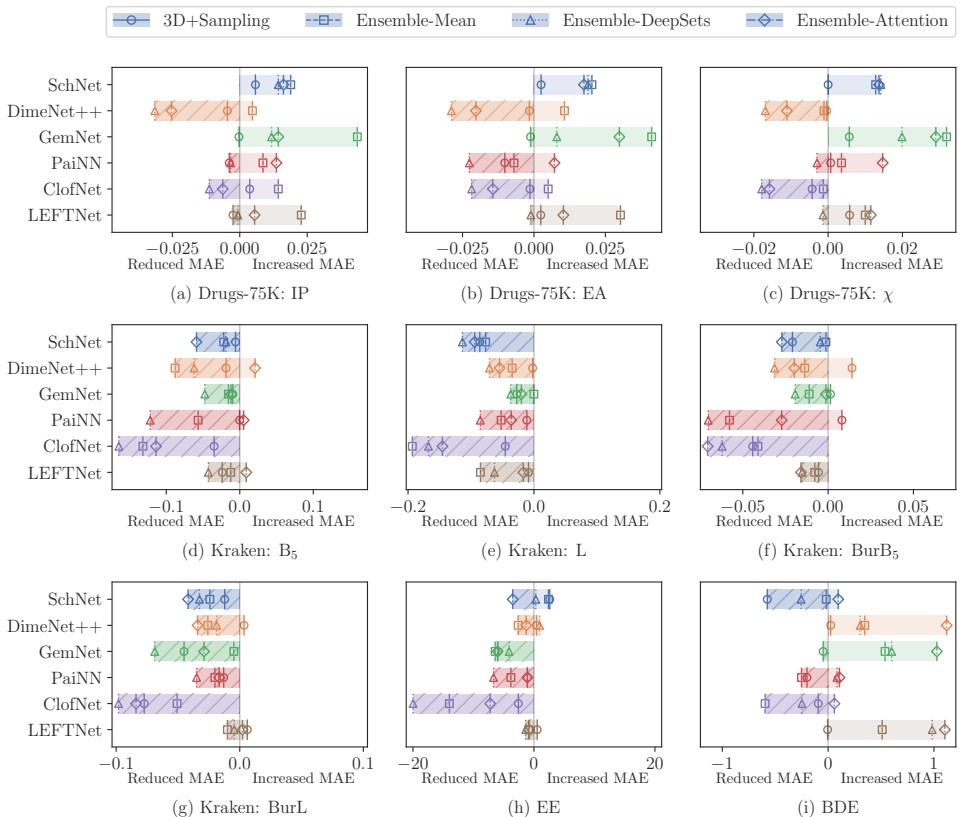

Figure 3: *Performance changes* of four conformer ensemble learning strategies on the basis of six 3D graph models. Here, negative values (marked in hatch patterns ) denote *reduced* Mean Absolute Error (MAE), signifying a performance improvement due to the incorporation of conformer ensembles.

because enantioselectivity depends on subtle spatial interactions. When predicting binding energies, using 3D models also leads to modest improvements.

Overall, model performance varies substantially across tasks, even within the same dataset, emphasizing the diversity of the tasks in MARCEL. Generally, 1D and 2D models perform well on small-scale molecular datasets, while 3D models excel on large datasets and reaction-centric tasks. MARCEL also highlights the benefits of explicitly encoding multiple conformers to improve MRL.

# 6  Discussions and Conclusions

In this work, we present the first MoleculAR Conformer Ensemble Learning benchmark (MARCEL) to evaluate the potential of learning from a set of conformer structures. Through two conformer ensemble learning strategies, we discover performance improvements across various tasks. However, there are some limitations that require further consideration. First, our studied ensemble learning strategies do not universally improve performance across all tasks and datasets. This highlights the need for more tailored approaches that integrate with domain expertise to better model specific tasks and datasets of practical interest. Second, the computational cost of encoding all conformers within the ensembles, especially for larger datasets, suggests the need to further study the trade-offs between model complexity and efficiency. Finally, our datasets only contain regression tasks and do not cover all of the relevant chemical space, which might limit the generalization of our experimental findings.

Despite these challenges, we envision that our work will prompt further research in the geometric deep learning community on how to make use of conformer ensembles for molecular property prediction. For instance, future research could explore new model architectures that can efficiently encode ensemble-level information or more sophisticated conformer sampling strategies. We also hope that our work will stimulate collaborative research across the machine learning and chemistry fields, with the ultimate goal of pushing the boundaries of predictive molecular modeling and aligning algorithmic advancements with the practical needs of the chemistry community.

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

# Supplementary Material for **MARCEL**

## A  Dataset Description

MARCEL include four datasets that cover a diverse range of chemical space, which focuses on four chemically-relevant tasks for both molecules and reactions, with an emphasis on Boltzmann-averaged properties of conformer ensembles computed at the Density-Functional Theory (DFT) level. Detailed information regarding dataset access, data formatting, and loading procedures can be found at our GitHub repository https://anonymous.4open.science/r/MARCEL-4813. Any subsequent updates will also be posted on this repository.

### A.1  Drugs-75K

Drugs-75K is a subset of the GEOM-Drugs [27] dataset, which includes 75,099 drug-like molecules with at least 5 rotatable bonds. The original GEOM-Drugs dataset was constructed using semi-empirical DFT methods, which is less accurate than full DFT. To curate the Drugs-75K subset, Auto3D [50] is used to generate and optimize the conformer ensembles for each molecule and AIMNet-NSE [51] is used to calculate three important DFT-based reactivity descriptors: ionization potential, electron affinity, and electronegativity [28].

Auto3D [50] efficiently generates high-quality conformers, with a mean RMSD at around 0.2 Å when compared with DFT conformers. It has been used in other large conformer dataset generation [52]. Regarding the neural network surrogate AIMNET-NSE [51], it mimics the PBE0/ma-def2-SVP method of DFT, which is widely used in the chemistry community. Investigating their accuracy is out of the scope of this paper, but are readily accessible from multiple sources [51, 53].

**Objectives.** The tasks are to predict the Boltzmann-averaged value of each property across the conformer ensemble $\langle y \rangle_{k_B} = \sum_{C_i \in \mathcal{C}} p_i y_i$, where $y_i$ is a conformer-specific property. We are given each $C_i$, and the goal is to predict $\langle y \rangle_{k_B}$ from the molecular graph G, a single conformer $C_i \in \mathcal{C}$, or the set $\mathcal{C}$.

**Dataset preparation.** In preparing the 75K version of GEOM-Drugs, we begin with the original SMILES strings of the molecules. We first exclude molecules that have less than 5 rotatable bonds. To enable the utilization of AIMNet-NSE for descriptor computation, we retain only those molecules containing atoms of H, C, N, O, F, Si, P, S, and Cl. Further, we generate DFT-level conformers and compute their energies with Auto3D. Based on these conformers, we compute three chemical bond energy descriptors using AIMNet-NSE. We exclude conformers that Auto3D fails to converge and charged molecules that are unable to be processed by AIMNet-NSE, which results in 75,099 molecules. Subsequently, we compute molecular-level Boltzmann-averaged descriptors based on

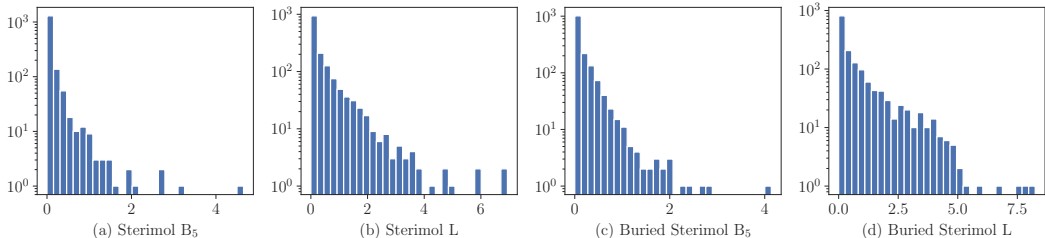

Figure S1: Histogram of the ratio of the variance of each conformer property to the variance of each Boltzmann-averaged property in the Kraken dataset.

conformer-level descriptors. Finally, we undertake a deduplication process as outlined in Section 3 with a RMSD threshold of 2.0, which yields a total of 558,002 distinct conformers.

**Data availability and license.** The original GEOM-Drugs dataset is publicly available at `https://github.com/learningmatter-mit/geom` but no license is specified. Our Drugs-75K can be accessed at `https://anonymous.4open.science/r/MARCEL-4813/datasets/Drugs/README.md`. As for the conformer ensembles and descriptors that we generated, they are licensed under the Apache License.

## A.2  Kraken

Kraken [29] is a dataset of 1,552 monodentate organophosphorus (III) ligands along with their DFT-computed conformer ensembles. In this study, we consider four 3D catalytic ligand descriptors exhibiting significant variance among conformers: Sterimol $B_5$, Sterimol L, buried Sterimol $B_5$, and buried Sterimol L. These descriptors quantify the steric size of a substituent in Å, and are commonly employed for Quantitative Structure-Activity Relationship (QSAR) modeling. The buried Sterimol variants describe the steric effects within the first coordination sphere of a metal [54].

**Objectives.** As in the Drugs-75K tasks, the goal is to predict the Boltzmann-averaged value of each property across the conformer ensemble from the molecular graph G, a single conformer $C_i \in \mathcal{C}$, or the set $\mathcal{C}$.

**Dataset preparation.** In this study, we utilize the original 3D geometry structures of molecules and their corresponding Boltzmann-averaged properties provided in the Kraken dataset. Among the 78 physical-organic properties listed in the original dataset, we select four properties that demonstrate high variance across conformer ensembles, as illustrated in Figure S1.

**Data availability and license.** The Kraken dataset is publicly accessible at `https://kraken.cs.toronto.edu`. Its copyright is retained by the original authors. Under the permission of the original authors, the Kraken dataset with the conformer ensembles and the four conformer-level descriptors used in this study can be accessed at `https://anonymous.4open.science/r/MARCEL-4813/datasets/Kraken/README.md`.

## A.3  EE

EE [30] is a dataset of 872 catalyst-substrate pairs involving 253 Rhodium (Rh)-bound atropisomeric catalysts derived from chiral bisphosphine, with 10 enamides as substrates. The dataset includes conformations of catalyst-substrate transition state complexes in two separate pro-S and pro-R configurations. The task is to predict the Enantiomeric Excess (EE) of the chemical reaction involving the substrate, defined as the absolute ratio between the concentration of each enantiomer in the product distribution.

**Objectives.** EE depends on the conformer ensembles of *each* pro-R and pro-S complex. The goal is to predict EE from the graphs of the catalyst and substrate ($G_{cat}$, $G_{sub}$), a conformer $C_i^{(R)} \in \mathcal{C}^{(R)}$ and $C_i^{(S)} \in \mathcal{C}^{(S)}$ for each complex, or the ensembles $\mathcal{C}^{(R)}$ and $\mathcal{C}^{(S)}$.

**Dataset preparation.** The conformer ensembles are generated with Q2MM, which automatically generates Transition State Force Fields (TSFFs) in order to simulate the conformer ensembles of each

prochiral transition state complex. Then, the EE values are computed from the conformer ensembles by Boltzmann-averaging the activation energies for the competing transition states [30, 55]. Finally, we conduct the same conformer deduplication process as described in Section 3 with a RMSD threshold of 1.0.

**Data availability and license.** As of now, the EE dataset is proprietary, given that the publication addressing the conformer ensembles is still under preparation. Therefore, access to the EE dataset is restricted to review purposes only. We anticipate making the EE dataset publicly accessible following the acceptance of the corresponding paper.

### A.4 BDE

BDE [31] is a dataset containing 5,915 organometallic catalysts $ML_1L_2$ consisting of a metal center (M = Pd, Pt, Au, Ag, Cu, Ni) coordinated to two flexible organic ligands ($L_1$ and $L_2$), each selected from a 91-membered ligand library. The data includes conformations of each unbound catalyst, as well as conformations of the catalyst when bound to ethylene and bromide after oxidative addition with vinyl bromide. Each catalyst has an electronic binding energy, computed as the difference in the minimum energies of the bound-catalyst complex and unbound catalyst, following the DFT-optimization of their respective conformer ensembles.

Although the binding energies are computed via DFT, the conformers provided for modeling are initially generated with Open Babel [32], followed by further geometric optimization steps, which ensures that the generated 3D structures are likely to be the global minimum energy conformers at the force field level [31, Supplementary Information]. We also note that obtaining DFT-optimized conformers for BDE is not feasible given the time-consuming nature of the process — a single geometric search using DFT can take 2 to 3 days. Therefore, this realistically represents the setting in which precise conformer ensembles are unknown at inference.

**Objectives.** The task is to predict the binding energy from the graphs of the unbound and bound catalyst, sampled conformers $C_i^{(\text{unbound})} \in \mathcal{C}^{(\text{unbound})}$ and $C_i^{(\text{bound})} \in \mathcal{C}^{(\text{bound})}$, or the ensembles $\mathcal{C}^{(\text{unbound})}$ and $\mathcal{C}^{(\text{bound})}$.

**Dataset preparation.** We employ Open Babel [32] to produce conformers for each unbound catalyst and each bound complex. In order to avoid redundancy, we follow a deduplication process as outlined in Section 3. For the unbound catalysts, a RMSD threshold value of 0.5 is applied, whereas for the bound complexes, a threshold of 1.0 is used.

**Data availability and license.** The binding energy descriptors can be accessed at `https://archive.materialscloud.org/record/2018.0014/v1` under the Creative Commons Attribution 4.0 International license. The conformers are publicly available at `https://anonymous.4open.science/r/MARCEL-4813/datasets/BDE/README.md` under the Apache license.

## B  Implementation Details

### B.1  Implementation of 1D Models

For the random forest model that operates on fingerprints, we employ three molecular fingerprint schemes: the Molecular ACCess System (MACCS) [37], Extended-Connectivity Fingerprints (ECFP) [36], and RDKit topological fingerprints [33]. Then, we concatenate their outputs into a single vector, which might lead to some feature redundancy, given the possible overlaps in these three fingerprint representations of the molecular structure. To tackle this issue, we remove any features that exhibit a high correlation exceeding 90% with the other features. For implementation, we employ Scikit-Learn [56] and compute fingerprints with RDKit [33].

For both LSTM and Transformer models that operate on SMILES strings, we use a Byte-Pair Encoding (BPE)-based tokenizer [57] that is pretrained on PubChem10M, which strikes a balance among character- and word-level representations and allows to handle large vocabularies in molecular corpora. For the Transformer model, we further follow the positional embedding scheme [39] to capture the positional relationship among tokens in the SMILES string.

Table S1: A summary of node and edge features used in 2D GNN models.

| | Feature | Explanation |
|---|---|---|
| | AtomicNum | Atomic number, representing the type of atom. |
| | ChiralTag | Indicator of chirality, a property of asymmetry. |
| | TotalDegree | Sum of implicit and explicit bonds of an atom. |
| | FormalCharge | Charge of an atom assuming equal sharing of bonding electrons. |
| Node | TotalNumHs | Total number of hydrogen atoms bonded to the atom. |
| | NumRadicalElectrons | Count of unpaired electrons in an atom. |
| | Hybridization | Type of atomic orbital hybridization in the atom. |
| | IsAromatic | Boolean indicating if the atom is part of an aromatic ring. |
| | IsInRing | Boolean indicating if the atom is part of any ring structure. |
| | BondType | Type of the bond (e.g., single, double, triple, aromatic). |
| Edge | Stereo | Stereochemistry of the bond (e.g., "none", "any", "Z", or "E" for double bonds). |
| | IsConjugated | Boolean indicating if the bond is part of a conjugated system. |

## B.2 Featurizations of Molecules for 2D Models

Following OGB [41], we employ a rich set of features for atoms (nodes) and bonds (edges) for 2D GNN models. A complete list of node and features can be found in Table S1.

## B.3 Hyperparameter Specifications and Experimental Environments

Each model is trained over 2,000 epochs using the Adam optimizer [49] with early stopping triggered if there is no improvement in the training loss over 200 epochs. To ensure a fair comparison, the hidden dimension size is uniformly set to 128 for all models. Other hyperparameters mostly follow the original configurations as described in the respective papers. The complete hyperparameter set of each model can be found in https://anonymous.4open.science/r/MARCEL-4813/benchmarks/params.

We utilize PyTorch [58] and PyTorch-Geometric [59] to implement all deep learning models. Most of the experiments are conducted on servers equipped with NVIDIA A100 GPUs, each with 40GB of memory. For memory-intensive models such as GemNet and LEFTNet, we use servers with NVIDIA H100 GPUs, each with 80GB memory. The cumulative computation time across all experiments amounts to approximately 6,000 single GPU hours.

## C  Additional Experiments on Evaluation Schemes of the Conformer Sampling Strategy

In this section, we conduct one additional experiment on the conformer ensemble learning strategies. We assess all 3D models on five tasks: Ionization Potential (IP) from the Drugs-75K dataset, $B_5$ and BurB$_5$ from the Kraken dataset, and tasks from the EE and BDE datasets.

In our previous setup, we evaluate the conformer sampling strategy using the lowest-energy conformer of each molecule at evaluation time, to provide a direct comparison to the single-conformer 3D models that are trained and tested with the lowest energy conformation. In these experiments, we continue to sample a random conformer uniformly from the conformer ensemble during training time, but consider two additional evaluation schemes: (1) evaluating model performance when encoding a randomly sampled conformer, and (2) evaluating model -performance when averaging the per-conformer predictions across the entire conformer ensemble.

The results of these experiments are summarized in Table S2. In the table, we refer to the original evaluation scheme as "fixed", and the additional schemes as "random" and "all", respectively. We find that across all three schemes, using the lowest-energy conformer for evaluation consistently yields the best performance. This is expected, as the lowest-energy conformer contributes the most to ensemble-level descriptors. The random conformer evaluation scheme generally yields the worst performance, which is likely due to the introduction of noise from less relevant conformers at test

Table S2: Performance comparison of three conformer sampling variants with different evaluation strategies. All models are trained with a randomly sampled conformer from the ensemble. The last column summarizes the average rank across all datasets for each base model.

| Model | Evaluation Strategy | Drugs-75K IP | Kraken B$_5$ | Kraken BurB$_5$ | EE | BDE | Average Rank |
|---|---|---|---|---|---|---|---|
| SchNet | Fixed | 0.4452 | 0.3235 | 0.2086 | 20.3595 | 1.9737 | 1 |
| | Random | 0.4498 | 0.3682 | 0.2454 | 22.0380 | 2.4416 | 3 |
| | All | 0.4428 | 0.3856 | 0.2407 | 18.0296 | 2.0106 | 2 |
| DimeNet++ | Fixed | 0.4395 | 0.3323 | 0.2237 | 15.0596 | 1.4741 | = 2 |
| | Random | 0.4555 | 0.3549 | 0.2222 | 13.5681 | 1.4688 | = 2 |
| | All | 0.4479 | 0.3282 | 0.2001 | 12.3562 | 1.6270 | 1 |
| GemNet | Fixed | 0.4066 | 0.2694 | 0.1796 | 12.0541 | 1.6059 | 1 |
| | Random | 0.4250 | 0.4034 | 0.2534 | 16.1709 | 1.7894 | 3 |
| | All | 0.4320 | 0.4523 | 0.2481 | 14.3952 | 1.6660 | 2 |
| PaiNN | Fixed | 0.4466 | 0.3441 | 0.2476 | 19.1521 | 1.9262 | 1 |
| | Random | 0.4770 | 0.3756 | 0.2478 | 21.3553 | 1.9411 | 3 |
| | All | 0.4478 | 0.3458 | 0.2342 | 19.1955 | 1.8696 | 2 |
| ClofNet | Fixed | 0.4430 | 0.4524 | 0.2442 | 31.3733 | 2.5126 | 1 |
| | Random | 0.4530 | 0.4689 | 0.2736 | 31.3675 | 2.6310 | = 2 |
| | All | 0.4363 | 0.4749 | 0.2855 | 34.3203 | 2.0271 | = 2 |
| LEFTNet | Fixed | 0.4149 | 0.2834 | 0.2120 | 20.3358 | 1.5276 | 1 |
| | Random | 0.4518 | 0.3177 | 0.2344 | 20.3740 | 1.5842 | 3 |
| | All | 0.4274 | 0.3152 | 0.2170 | 18.8945 | 1.8663 | 2 |

time. Interestingly, we observe occasional performance improvement when averaging the predictions across all conformers in the ensemble, indicating that explicitly using ensemble-level information during evaluation can be beneficial.

## D Raw Data

The raw performance data with standard deviation of Table 2 and Figure 3 is summarized in Table S3.

Table S3: Raw performance data (mean ± standard deviation) of representative 1D, 2D, 3D, and conformer ensemble MRL models in terms of absolute test error.

| Category | Model | | Drugs-75K | | | Kraken | | | | EE | BDE |
|---|---|---|---|---|---|---|---|---|---|---|---|
| | | | IP | EA | $\chi$ | $B_5$ | L | $BurB_5$ | BurL | | |
| 1D | Random forest | | $0.4987_{\pm0.0037}$ | $0.4747_{\pm0.0022}$ | $0.2732_{\pm0.0031}$ | $0.4760_{\pm0.0041}$ | $0.4303_{\pm0.0090}$ | $0.2758_{\pm0.0180}$ | $0.1521_{\pm0.0149}$ | $61.2963_{\pm2.8640}$ | $3.0335_{\pm0.2693}$ |
| | LSTM | | $0.4788_{\pm0.0024}$ | $0.4648_{\pm0.0002}$ | $0.2505_{\pm0.0050}$ | $0.4879_{\pm0.0280}$ | $0.5142_{\pm0.0411}$ | $0.2813_{\pm0.0041}$ | $0.1924_{\pm0.0028}$ | $64.0088_{\pm2.3708}$ | $2.8279_{\pm0.0728}$ |
| | Transformer | | $0.6617_{\pm0.0023}$ | $0.5850_{\pm0.0031}$ | $0.4073_{\pm0.0006}$ | $0.9611_{\pm0.0813}$ | $0.8389_{\pm0.0431}$ | $0.4929_{\pm0.0369}$ | $0.2781_{\pm0.0207}$ | $62.0816_{\pm2.1789}$ | $10.0771_{\pm0.6457}$ |
| 2D | GIN | | $0.4354_{\pm0.0029}$ | $0.4169_{\pm0.0032}$ | $0.2260_{\pm0.0017}$ | $0.3128_{\pm0.0264}$ | $0.4003_{\pm0.0341}$ | $0.1719_{\pm0.0031}$ | $0.1200_{\pm0.0040}$ | $62.3065_{\pm2.9010}$ | $2.6368_{\pm0.2276}$ |
| | GIN-VN | | $0.4361_{\pm0.0059}$ | $0.4169_{\pm0.0083}$ | $0.2267_{\pm0.0002}$ | $0.3567_{\pm0.0031}$ | $0.4344_{\pm0.0416}$ | $0.2422_{\pm0.0033}$ | $0.1741_{\pm0.0109}$ | $62.3815_{\pm2.1882}$ | $2.7417_{\pm0.2446}$ |
| | ChemProp | | $0.4595_{\pm0.0028}$ | $0.4417_{\pm0.0045}$ | $0.2441_{\pm0.0012}$ | $0.4850_{\pm0.0068}$ | $0.5452_{\pm0.0454}$ | $0.3002_{\pm0.0086}$ | $0.1948_{\pm0.0138}$ | $61.0336_{\pm2.9715}$ | $2.6616_{\pm0.1429}$ |
| | GraphGPS | | $0.4351_{\pm0.0049}$ | $0.4085_{\pm0.0055}$ | $0.2212_{\pm0.0054}$ | $0.3450_{\pm0.0324}$ | $0.4363_{\pm0.0133}$ | $0.2066_{\pm0.0115}$ | $0.1500_{\pm0.0138}$ | $61.6251_{\pm1.3743}$ | $2.4827_{\pm0.1992}$ |
| 3D | SchNet | | $0.4394_{\pm0.0062}$ | $0.4207_{\pm0.0021}$ | $0.2243_{\pm0.0089}$ | $0.3293_{\pm0.0068}$ | $0.5458_{\pm0.0341}$ | $0.2295_{\pm0.0111}$ | $0.1861_{\pm0.0095}$ | $17.7421_{\pm1.0899}$ | $2.5488_{\pm0.0050}$ |
| | DimeNet++ | | $0.4441_{\pm0.0087}$ | $0.4233_{\pm0.0072}$ | $0.2436_{\pm0.0075}$ | $0.3510_{\pm0.0107}$ | $0.4174_{\pm0.0397}$ | $0.2097_{\pm0.0160}$ | $0.1526_{\pm0.0072}$ | $14.6414_{\pm2.2791}$ | $1.4503_{\pm0.0370}$ |
| | GemNet | | $0.4069_{\pm0.0007}$ | $0.3922_{\pm0.0024}$ | $0.1970_{\pm0.0039}$ | $0.2789_{\pm0.0125}$ | $0.3754_{\pm0.0086}$ | $0.1782_{\pm0.0099}$ | $0.1635_{\pm0.0063}$ | $18.0338_{\pm2.4777}$ | $1.6530_{\pm0.3081}$ |
| | PaiNN | | $0.4505_{\pm0.0041}$ | $0.4495_{\pm0.0054}$ | $0.2324_{\pm0.0040}$ | $0.3443_{\pm0.0388}$ | $0.4471_{\pm0.0324}$ | $0.2395_{\pm0.0176}$ | $0.1673_{\pm0.0088}$ | $20.2359_{\pm1.2128}$ | $2.1261_{\pm0.0920}$ |
| | ClofNet | | $0.4393_{\pm0.0084}$ | $0.4251_{\pm0.0066}$ | $0.2378_{\pm0.0020}$ | $0.4873_{\pm0.0093}$ | $0.6417_{\pm0.0362}$ | $0.2884_{\pm0.0166}$ | $0.2529_{\pm0.0052}$ | $33.9473_{\pm1.4633}$ | $2.6057_{\pm0.0236}$ |
| | LEFTNet | | $0.4174_{\pm0.0009}$ | $0.3964_{\pm0.0009}$ | $0.2083_{\pm0.0054}$ | $0.3072_{\pm0.0012}$ | $0.4493_{\pm0.0261}$ | $0.2176_{\pm0.0010}$ | $0.1486_{\pm0.0095}$ | $19.7974_{\pm1.4097}$ | $1.5328_{\pm0.0567}$ |
| 3D +Sampling | SchNet | | $0.4452_{\pm0.0080}$ | $0.4232_{\pm0.0042}$ | $0.2243_{\pm0.0022}$ | $0.3235_{\pm0.0147}$ | $0.4598_{\pm0.0041}$ | $0.2086_{\pm0.0111}$ | $0.1739_{\pm0.0142}$ | $20.3595_{\pm1.5260}$ | $1.9737_{\pm0.0125}$ |
| | DimeNet++ | | $0.4395_{\pm0.0032}$ | $0.4217_{\pm0.0040}$ | $0.2432_{\pm0.0048}$ | $0.3323_{\pm0.0320}$ | $0.4153_{\pm0.0208}$ | $0.2237_{\pm0.0122}$ | $0.1561_{\pm0.0241}$ | $15.0596_{\pm0.2867}$ | $1.4741_{\pm0.0349}$ |
| | GemNet | | $0.4066_{\pm0.0015}$ | $0.3910_{\pm0.0004}$ | $0.2027_{\pm0.0013}$ | $0.2694_{\pm0.0221}$ | $0.3488_{\pm0.0252}$ | $0.1796_{\pm0.0098}$ | $0.1184_{\pm0.0033}$ | $12.0541_{\pm0.7735}$ | $1.6059_{\pm0.1094}$ |
| | PaiNN | | $0.4466_{\pm0.0087}$ | $0.4393_{\pm0.0045}$ | $0.2331_{\pm0.0037}$ | $0.3441_{\pm0.0161}$ | $0.4358_{\pm0.0343}$ | $0.2476_{\pm0.0070}$ | $0.1543_{\pm0.0022}$ | $19.1521_{\pm0.2386}$ | $1.9262_{\pm0.0188}$ |
| | ClofNet | | $0.4430_{\pm0.0074}$ | $0.4237_{\pm0.0005}$ | $0.2335_{\pm0.0090}$ | $0.4524_{\pm0.0935}$ | $0.5962_{\pm0.0074}$ | $0.2442_{\pm0.0109}$ | $0.1756_{\pm0.0112}$ | $31.3733_{\pm1.9892}$ | $2.5126_{\pm0.2366}$ |
| | LEFTNet | | $0.4149_{\pm0.0019}$ | $0.3988_{\pm0.0048}$ | $0.2141_{\pm0.0084}$ | $0.2834_{\pm0.0068}$ | $0.4407_{\pm0.0531}$ | $0.2120_{\pm0.0097}$ | $0.1547_{\pm0.0101}$ | $20.3358_{\pm0.6614}$ | $1.5276_{\pm0.0088}$ |
| Ensemble | SchNet | Mean | $0.4583_{\pm0.0019}$ | $0.4410_{\pm0.0018}$ | $0.2371_{\pm0.0098}$ | $0.3075_{\pm0.0151}$ | $0.4691_{\pm0.0234}$ | $0.2282_{\pm0.0206}$ | $0.1619_{\pm0.0062}$ | $20.1392_{\pm1.5748}$ | $2.5312_{\pm0.0246}$ |
| | | DeepSet | $0.4537_{\pm0.0065}$ | $0.4396_{\pm0.0010}$ | $0.2385_{\pm0.0066}$ | $0.3105_{\pm0.0381}$ | $0.4322_{\pm0.0464}$ | $0.2249_{\pm0.0234}$ | $0.1535_{\pm0.0076}$ | $18.0495_{\pm1.2846}$ | $2.2941_{\pm0.2229}$ |
| | | Attention | $0.4556_{\pm0.0075}$ | $0.4382_{\pm0.0125}$ | $0.2380_{\pm0.0007}$ | $0.2704_{\pm0.0187}$ | $0.4517_{\pm0.0132}$ | $0.2024_{\pm0.0183}$ | $0.1443_{\pm0.0043}$ | $14.2238_{\pm0.5451}$ | $2.6445_{\pm0.0031}$ |
| | DimeNet++ | Mean | $0.4488_{\pm0.0086}$ | $0.4340_{\pm0.0079}$ | $0.2425_{\pm0.0060}$ | $0.2630_{\pm0.0122}$ | $0.3828_{\pm0.0331}$ | $0.1960_{\pm0.0059}$ | $0.1268_{\pm0.0060}$ | $12.0259_{\pm0.8933}$ | $1.7964_{\pm0.1260}$ |
| | | DeepSet | $0.4126_{\pm0.0076}$ | $0.3944_{\pm0.0034}$ | $0.2267_{\pm0.0047}$ | $0.2889_{\pm0.0069}$ | $0.3468_{\pm0.0090}$ | $0.1783_{\pm0.0110}$ | $0.1339_{\pm0.0087}$ | $15.5754_{\pm2.6294}$ | $1.7533_{\pm0.0163}$ |
| | | Attention | $0.4188_{\pm0.0024}$ | $0.4030_{\pm0.0075}$ | $0.2325_{\pm0.0028}$ | $0.3718_{\pm0.0300}$ | $0.3628_{\pm0.0259}$ | $0.1899_{\pm0.0081}$ | $0.1185_{\pm0.0105}$ | $13.3643_{\pm1.4309}$ | $2.5714_{\pm0.2149}$ |
| | GemNet | Mean | $0.4505_{\pm0.0052}$ | $0.4334_{\pm0.0023}$ | $0.2289_{\pm0.0032}$ | $0.2635_{\pm0.0053}$ | $0.3753_{\pm0.0036}$ | $0.1671_{\pm0.0154}$ | $0.1587_{\pm0.0029}$ | $11.6142_{\pm1.7271}$ | $2.1914_{\pm0.0605}$ |
| | | DeepSet | $0.4187_{\pm0.0022}$ | $0.4002_{\pm0.0012}$ | $0.2169_{\pm0.0036}$ | $0.2313_{\pm0.0026}$ | $0.3386_{\pm0.0269}$ | $0.1589_{\pm0.0068}$ | $0.0947_{\pm0.0012}$ | $13.9273_{\pm1.8656}$ | $2.2532_{\pm0.2106}$ |
| | | Attention | $0.4212_{\pm0.0017}$ | $0.4221_{\pm0.0097}$ | $0.2260_{\pm0.0056}$ | $0.2670_{\pm0.0026}$ | $0.3554_{\pm0.0147}$ | $0.1769_{\pm0.0153}$ | $0.1346_{\pm0.0075}$ | $12.0249_{\pm1.8418}$ | $2.6810_{\pm0.0223}$ |
| | PaiNN | Mean | $0.4591_{\pm0.0024}$ | $0.4425_{\pm0.0064}$ | $0.2360_{\pm0.0032}$ | $0.2877_{\pm0.0252}$ | $0.3950_{\pm0.0233}$ | $0.1817_{\pm0.0091}$ | $0.1472_{\pm0.0039}$ | $16.4239_{\pm0.0743}$ | $1.8744_{\pm0.1657}$ |
| | | DeepSet | $0.4471_{\pm0.0071}$ | $0.4269_{\pm0.0033}$ | $0.2294_{\pm0.0065}$ | $0.2225_{\pm0.0218}$ | $0.3619_{\pm0.0192}$ | $0.1693_{\pm0.0111}$ | $0.1324_{\pm0.0091}$ | $13.5570_{\pm0.5505}$ | $2.2097_{\pm0.0586}$ |
| | | Attention | $0.4641_{\pm0.0016}$ | $0.4567_{\pm0.0094}$ | $0.2471_{\pm0.0049}$ | $0.3496_{\pm0.0140}$ | $0.4109_{\pm0.0167}$ | $0.2123_{\pm0.0005}$ | $0.1506_{\pm0.0029}$ | $19.1556_{\pm2.2765}$ | $2.2335_{\pm0.1255}$ |
| | ClofNet | Mean | $0.4536_{\pm0.0030}$ | $0.4301_{\pm0.0007}$ | $0.2365_{\pm0.0075}$ | $0.3555_{\pm0.0193}$ | $0.4485_{\pm0.0053}$ | $0.2473_{\pm0.0076}$ | $0.2022_{\pm0.0212}$ | $19.9710_{\pm0.7745}$ | $2.0106_{\pm0.0856}$ |
| | | DeepSet | $0.4280_{\pm0.0056}$ | $0.4033_{\pm0.0024}$ | $0.2199_{\pm0.0073}$ | $0.3228_{\pm0.0020}$ | $0.4742_{\pm0.0161}$ | $0.2263_{\pm0.0249}$ | $0.1548_{\pm0.0039}$ | $13.9647_{\pm1.2753}$ | $2.3576_{\pm0.0496}$ |
| | | Attention | $0.4330_{\pm0.0071}$ | $0.4107_{\pm0.0048}$ | $0.2220_{\pm0.0084}$ | $0.3734_{\pm0.0267}$ | $0.4963_{\pm0.0286}$ | $0.2178_{\pm0.0186}$ | $0.1690_{\pm0.0281}$ | $26.7133_{\pm1.7225}$ | $2.6652_{\pm0.1438}$ |
| | LEFTNet | Mean | $0.4402_{\pm0.0062}$ | $0.4267_{\pm0.0026}$ | $0.2183_{\pm0.0007}$ | $0.2949_{\pm0.0001}$ | $0.3643_{\pm0.0352}$ | $0.2098_{\pm0.0146}$ | $0.1386_{\pm0.0007}$ | $18.9245_{\pm2.0136}$ | $2.0440_{\pm0.0076}$ |
| | | DeepSet | $0.4167_{\pm0.0043}$ | $0.3953_{\pm0.0000}$ | $0.2069_{\pm0.0022}$ | $0.2644_{\pm0.0130}$ | $0.3866_{\pm0.0270}$ | $0.2023_{\pm0.0026}$ | $0.1441_{\pm0.0042}$ | $18.4189_{\pm1.8922}$ | $2.5165_{\pm0.3077}$ |
| | | Attention | $0.4229_{\pm0.0059}$ | $0.4067_{\pm0.0047}$ | $0.2198_{\pm0.0011}$ | $0.3161_{\pm0.0116}$ | $0.4324_{\pm0.0292}$ | $0.2017_{\pm0.0023}$ | $0.1508_{\pm0.0075}$ | $18.9988_{\pm1.6904}$ | $2.6361_{\pm0.1560}$ |

