# OpenReview forum: "Learning Over Molecular Conformer Ensembles: Datasets and Benchmarks"
_NeurIPS.cc/2023/Workshop/AI4Science — NeurIPS2023-AI4Science Poster_

### Official Review · Reviewer_zYqu · 2023-10-23
**A promising framework, not enough domain knowledge**

**Rating:** 6
**Confidence:** 5

**Review:**

Strengths:

-	The authors propose a research problem that is closely related to the NeurIPS community and, in general, structural research of molecules.

-	It is an interesting and very promising direction, with great interest across many disjoint fields.

-	The paper is well-structured and easy to understand for scientists.

-	The methodology is explained well and supported by efficient results.

-	The authors’ work (MARCEL) could expedite the ability of researchers to gauge the performance of both new and existing machine learning models trained on conformer datasets; specifically as it relates to input structure, training protocol and model architecture.

-	If maintained, the authors’ framework could be extended as the field advances, with potential for many of the below weaknesses to be accounted for.

Weaknesses:

-	Although the authors admit the need for integrating domain expertise in their Conclusions, it is alarming as a reader to be presented so little of it throughout the main text. Specifically, the authors are in general too vague on the precise expectations of conformers in thermodynamic equilibrium at different molecular scales. Addressing the following concerns may help shield this work from contention or rebuke by scientists who study conformers, for which the authors’ method is intended.

-	First, the authors should readily describe the size of their systems being considered. What is the range of molecular weights or atoms in each structure that the authors are interested in studying? Since different scales offer substantially different types of trajectories between conformational states (e.g., multidimensional, linear, branching or cyclic), the authors should describe the regimes in which they anticipate their methods will be most and least applicable. This is important since the training of models to predict conformational properties is at the forefront of interest in prominent fields such as cryo-EM, which typically deal with large-scale complexes such as ATP synthase (40,000 atoms) and the Ribosome (300,000 atoms).

-	To ease frustration with this point, this reader took on the assumption shortly into the authors’ paper that the authors had no interest in describing or modeling the conformational properties of large-scale macromolecular complexes (for which there is bountiful research regarding their conformational continuum in the community; e.g., the work of Nobel laureate Joachim Frank; and in ML: H. Gupta 2021; E. Zhong 2021, etc.). At this scale, complexes undergo large, orchestrated conformational shifts well beyond small rotations and “minor vibrational perturbations”, dependent on complex free-energy landscape trajectories with multiple isolated minima which cannot be accounted for using local perturbation simulations or interpolations of the lowest-energy state. Thus, it is this reviewer’s expectation that the authors’ proposed framework will be inadequate in this regime. (As a note on scale, the number of atoms advised for the GFN2-xTB method used in the authors’ GEOM-Drugs-based dataset is ~1,000).

-	Whether or not the conformational spectrum of a (much) smaller system can be adequately modeled using the proposed methods is another question worth asking. In general, there is no rationale provided that the authors’ chosen conformer datasets have been generatively sampled from the entirety of their expected in vivo-accessible free-energy landscapes, or at least from a large enough region of that space to constitute novel progression along a metabolic trajectory (and not just a jiggling about in one free-energy basin). Indeed, one would expect that the performance of the authors’ applied models would be heavily influenced by the distribution of conformers sampled from the conformational state space, and thus the degrees of freedom defining each coordinate of that space and its span. However, there is no mention of this relationship. Instead, the authors attribute all (prominently) observed inconsistency in model performance to “computational burden”.

-	As to another point, the authors should distinguish between supervised and unsupervised approaches in this field, which would ultimately include a discussion of existing embedding strategies using linear and nonlinear dimensionality reduction. This is a key point since the authors prescribe a relatively plain method in Strategy 2 for embedding conformers, which is already a widely explored space in the molecular dynamics and cryo-EM community, and entire papers have been written there relating the space of naturally and synthetically embedded conformers to their conformational and free-energy properties (e.g., A. Ferguson 2010; E. Seitz, …, J. Frank 2022).

-	The novelty of the work is also difficult to assess, since so much of it is derived from existing databases and models. It appears that the authors have cleaned up larger datasets of existing 3D conformers and consolidated several models into one package (MARCEL), which is not a substantial novelty that the abstract should lean on. The authors’ design choices for processing these datasets and their heuristic study appears to be the main novelty, but this reviewer is too uninformed in that literature to gauge it. To further the novelty and scientific impact, will the authors make MARCEL available in open-source code or comprehensive GUI for the community? Similarly, will there be an accompanying user manual and tutorials for its use, etc.?

---

### Official Review · Reviewer_x1gu · 2023-10-24
**A well motivated benchmark for learning from 3D ensembles, more models can be included**

**Rating:** 7
**Confidence:** 4

**Review:**

## Paper Summary

The paper benchmarks molecular predictive models for their ability to infer form 3D conformer ensembles. The paper benchmarks several architectures that operate on 1D, 2D or 3D molecular data.

## Strengths

- The benchmark is well motivated. Although there was significant attention on 3D molecule conformer generation models in the recent years, inference from 3D conformer ensembles is a less studied in comparison.
- The benchmark design is interesting, using different architectures for 1D, 2D and 3D molecular data. The datasets are diverse.
- Observations and discussions are interesting.

## Weeknesses

- I would suggest the authors to elaborate on related work on the problem of inferring from 3D conformer ensembles.
- I would suggest the authors to include more architectures that can explicitly 3D conformer ensembles. For example, the models that proposed specifically to operate on 3D conformer ensembles, such as [1], or the models that learn implicit representation from 2D and 3D graphs, such as [2].
- While Drugs 75k is a relatively well studied dataset, the other datasets are smaller and unknown in the ML communties. I would suggest the authors to  include more information about learning from these datasets such as significance of the tasks, ability for generalization and performance limits.

[1] Axelrod, Simon, and Rafael Gomez-Bombarelli. "Molecular machine learning with conformer ensembles." *Machine Learning: Science and Technology* 4.3 (2023): 035025.

[2] Stärk, Hannes, et al. "3d infomax improves gnns for molecular property prediction." *International Conference on Machine Learning*. PMLR, 2022.